# Colon adenocarcinoma-derived cells possessing stem cell function can be modulated using renin-angiotensin system inhibitors

Matthew J. Munro[1,2], Lifeng Peng[2], Susrutha K. Wickremesekera[1,3], Swee T. Tan[1,4,5]*

1 Gillies McIndoe Research Institute, Wellington, New Zealand, 2 School of Biological Sciences and Centre for Biodiscovery, Victoria University of Wellington, Kelburn, Wellington, New Zealand, 3 Upper Gastrointestinal, Hepatobiliary & Pancreatic Section, Department of General Surgery, Wellington Regional Hospital, Wellington, New Zealand, 4 Wellington Regional Plastic, Maxillofacial & Burns Unit, Hutt Hospital, Wellington, New Zealand, 5 Department of Surgery, The Royal Melbourne Hospital, The University of Melbourne, Melbourne, Victoria, Australia

* swee.tan@gmri.org.nz

**Data Availability Statement:** The data underlying the results presented in the study are available from https://osf.io/ukd9b/.

## Abstract

The cancer stem cell (CSC) concept proposes that cancer recurrence and metastasis are driven by CSCs. In this study, we investigated whether cells from colon adenocarcinoma (CA) with a CSC-like phenotype express renin-angiotensin system (RAS) components, and the effect of RAS inhibitors on CA-derived primary cell lines. Expression of RAS components was interrogated using immunohistochemical and immunofluorescence staining in 6 low-grade CA (LGCA) and 6 high-grade CA (HGCA) tissue samples and patient-matched normal colon samples. Primary cell lines derived from 4 HGCA tissues were treated with RAS inhibitors to investigate their effect on cellular metabolism, tumorsphere formation and transcription of pluripotency genes. Immunohistochemical and immunofluorescence staining showed expression of $AT_2R$, ACE2, PRR, and cathepsins B and D by cells expressing pluripotency markers. β-blockers and $AT_2R$ antagonists reduced cellular metabolism, pluripotency marker expression, and tumorsphere-forming capacity of CA-derived primary cell lines. This study suggests that the RAS is active in CSC-like cells in CA, and further investigation is warranted to determine whether RAS inhibition is a viable method of targeting CSCs.

## Introduction

The cancer stem cell (CSC) concept hypothesizes that tumor growth is driven by CSCs, a small subpopulation of cancer cells with stem cell characteristics [1–4]. CSCs divide asymmetrically to produce identical CSCs, as well as progenitor cells–the transit amplifying cells of the colon [5–7]. CSCs have lost control of cellular homeostasis leading to dysregulated cellular replication and differentiation, resulting in uncontrolled growth and tumor formation [1–3, 8]. CSCs resist conventional therapies and are responsible for tumor recurrence and metastasis [1–3].

**Funding:** This project was supported by Lloyd Morrison Trust in the form of research funding awarded to STT and New Zealand Community Trust in the form of a PhD Scholarship awarded to MJM. The funders had no role in the study design, data collection and analysis, decision to publish, or preparation of the manuscript.

**Competing interests:** The authors have read the journal's policy and have the following competing interests: STT is the inventor of the patents Cancer Diagnosis and Therapy (PCT/NZ2015/050108), Cancer Therapeutic (PCT/NZ2018/050006), Novel Pharmaceutical Compositions for Cancer Therapy (US/62/711709), and Cancer diagnosis and therapy (United States Patent No. 10281472). This does not alter our adherence to PLOS ONE policies on sharing data and materials. There are no other patents, products in development or marketed products associated with this research to declare.

The renin-angiotensin system (RAS) maintains blood pressure and fluid volume by controlling sodium absorption, vascular tone and hormone release [9], and its involvement in cancer is an emerging field of research, particularly its paracrine role in regulating CSCs [10]. The classical RAS (S1 Fig) begins with renin, produced as the pro-enzyme pro-renin by the juxtaglomerular cells of the kidney [11], and is activated by proteolytic cleavage [9]. Active renin converts angiotensinogen (AGT) to angiotensin I (ATI), which is converted by angiotensin-converting enzyme (ACE) into angiotensin II (ATII) [11, 12]. ACE is expressed on the endothelium of blood vessels and circulates in the plasma [11]. ATII is the major effector molecule of the RAS via ATII receptor 1 ($AT_1R$) and ATII receptor 2 ($AT_2R$) [11, 12]. ACE2 converts ATI into Angiotensin1-9 (Ang1-9) and Angiotensin1-7 (Ang1-7), and ATII into Ang1-7, which bind the Mas receptor (MasR). This antagonizes the effects of $AT_1R$-mediated signaling, similar to that of the ATII/$AT_2R$ axis [9], by reducing inflammation, oxidative stress and susceptibility to cardiovascular diseases [11].

The (pro)renin receptor (PRR) is a 35 kDa receptor with no intrinsic kinase ability [13]. It is a single-pass transmembrane protein that is cleaved to form soluble PRR (sPRR) with a molecular weight of ~28 kDa [13]. Full-length PRR and sPRR can bind pro-renin and mature renin. Pro-renin is usually activated by irreversible proteolytic removal of a 43 amino acid pro-segment by pepsin [13]. When bound to PRR, pro-renin becomes reversibly active without proteolysis via a conformational change [13], allowing pro-renin to cleave AGT to produce ATI with four times greater activity [13]. PRR was discovered to be a vital component of the Wnt signaling receptor complex during investigation of proteins that co-purify with the hydrogen ion pump V-ATPase [13].

The RAS is involved in cancer-related processes such as angiogenesis, proliferation, tumorigenesis and metastasis, with ATII the main driver [10, 14]. However, the downstream effects of $AT_1R$ and $AT_2R$ play antagonistic roles. $AT_1R$ is associated with adverse or cancer-related outcomes through the actions of VEGF, PDGF and FGF, in processes such as angiogenesis, proliferation, inflammation and fibrosis, while $AT_2R$ antagonizes these effects [10, 15, 16]. $AT_1R$ and $AT_2R$ are expressed in the plasma membrane and $AT_2R$ in the nucleus of colorectal cancer (CRC) cells, in which ATII binding to $AT_1R$ leads to tumor growth and invasion and VEGF-A secretion, and $AT_2R$ opposes all actions of $AT_1R$ at high levels of ATII [17].

Cathepsins are lysosomal peptidases belonging to the papain family [18, 19]. Cathepsins B, D and G constitute bypass loops of the RAS [10] (S1 Fig).

RAS components are targeted by three common classes of drugs in hypertension treatment: β-blockers, which reduce renin production; ACE inhibitors (ACEIs); and angiotensin receptor blockers (ARBs), which antagonize $AT_1R$. Retrospective epidemiological studies show that patients taking these medications have lower incidence of polyp formation, CRC and metastasis, with ARBs having the greatest effect [16]. A large meta-analysis showed a positive trend in survival associated with ARBs and ACEIs in CRC patients, with up to a 20% increase in disease-free survival, progression-free survival and overall survival [20].

β-blockers are antagonists of adrenergic receptors, which bind epinephrine and norepinephrine as part of the sympathetic nervous system [21]. β-blockers restrict prolonged adrenergic stimulation and subsequent renin production [11]. Captopril, the first orally active ACEI developed, has high specificity for ACE [22]. Conversely, cilazapril is administered as a prodrug and is metabolized to cilazaprilat [23] which is more potent than captopril and has better absorption following oral administration [23]. ARBs are $AT_1R$ antagonists that bind competitively over ATII and have a slow dissociation from $AT_1R$ [24]. Currently, there are no $AT_2R$ antagonists in clinical use. However, one candidate, EMA401, has undergone multiple phase I clinical trials to demonstrate its tolerability, and a phase II trial for treating neuropathic pain [25, 26].

We have previously identified CSC-like cells in CA using a gene panel consisting of *OCT4*, *SOX2*, *NANOG*, *KLF4* and *c-MYC*, which regulate pluripotency and are used to produce induced pluripotent stem cells [27, 28]. The extent and localization of their expression differs between CA and normal colon (NC) tissues. Except for OCT4, these markers are co-expressed by a small subpopulation of cells in the epithelium that also expresses CD133 and LGR5. OCT4 is expressed by elongated cells within the stroma of CA but not NC tissue. The markers display distinct expression profiles in primary cell lines derived from low-grade CA (LGCA) and high-grade CA (HGCA) tissues, which are capable of forming tumorspheres that can be recapitulated upon passaging and can differentiate into different lineages.

We have demonstrated co-localization of RAS components and stemness-associated markers in glioblastoma [29], renal clear cell carcinoma [30], oral cavity squamous cell carcinoma (SCC) [31, 32], primary [33] and metastatic [34] cutaneous SCC, metastatic malignant melanoma [35, 36] and CA liver metastases [37], but not yet in primary CA. We hypothesized that CSC-like cells in CA expressed RAS components. This study investigated the expression and localization of RAS components and cathepsin B (CTSB), cathepsin D (CTSD) and cathepsin G (CTSG), in CA tissue samples, CA tissue-derived primary cell lines, and patient-matched NC tissues. We also explored the effects of RAS inhibitors (RASIs) on mRNA expression of stemness-associated markers, cellular metabolism, and tumorsphere forming capacity of CA-derived primary cell lines.

## Materials and methods

### Tissue samples

Formalin-fixed paraffin-embedded (FFPE) and snap-frozen tissue samples from 6 LGCA and 6 HGCA patients with patient-matched NC tissues were provided by the Gillies McIndoe Research Institute Tissue Bank (GMRITB) for this study, approved by the Central Health and Disability Ethics Committee (Ref. 15/CEN/106) with written informed consent from all participants.

### Immunohistochemical staining

Immunohistochemical (IHC) staining was performed on 4 μm sections of all formalin-fixed paraffin embedded (FFPE) tissue samples. Pre-defined automated staining protocols were carried out using the Leica BOND™ RX auto-stainer (Leica, Nussloch, Germany), with 3,3'-diaminobenzidine as the chromogen. Primary antibodies used are outlined in Table 1.

### Immunofluorescence staining

Protein co-localization was performed using dual immunofluorescence (IF) staining with the same primary antibodies used for IHC staining. Secondary antibodies used were Vectafluor Excel goat anti-mouse 488 (ready-to-use; cat# DK2488, Vector Laboratories, Burlingame, CA, USA) and Alexa Fluor donkey anti-rabbit 594 (1:500; cat# ab150076, Life Technologies, Carlsbad, CA, USA). All stained slides were mounted using Vecta Shield Hardset mounting medium with 4',6-diamino-2-phenylindole nuclear stain (cat# H-1200, Vector, Abacus DX, Auckland, New Zealand). Negative controls were performed using matched isotype controls for both mouse (ready-to-use; cat# IR750, Dako, Copenhagen, Denmark) and rabbit (ready-to-use; cat# IR600, Dako).

### Image capture and analysis

Images of IHC-stained slides were captured using an Olympus BX53 light microscope and an Olympus SC100 digital camera (Olympus, Tokyo, Japan). IF-stained slides were visualized and

**Table 1. Primary antibodies used for immunohistochemical and immunofluorescence staining.**

| Marker | Species/clonality | Dilution | Catalogue number |
|---|---|---|---|
| OCT4 | Mouse monoclonal | 1:30 | MRQ-10 (Cell Marque) |
| SOX2 | Rabbit polyclonal | 1:200 | ab97959 (Abcam) |
| NANOG | Rabbit monoclonal | 1:200 | 443R-16 (Cell Marque) |
| NANOG | Mouse monoclonal | 1:100 | ab62734 (Abcam) |
| KLF4 | Rabbit polyclonal | 1:200 | NBP2-24749SS (Novus) |
| c-MYC | Rabbit monoclonal | 1:100 | ab32072 (Abcam) |
| PRR | Rabbit polyclonal | 1:200 | ab264763 (Abcam) |
| ACE | Rabbit polyclonal | 1:50 | PA5-83080 (ThermoFisher Scientific) |
| ACE2 | Mouse monoclonal | 1:1000 | MAB933 (ThermoFisher Scientific) |
| AT$_2$R | Rabbit polyclonal | 1:2000 | NBP1-77368 (Novus) |
| CTSB | Mouse monoclonal | 1:200 | ab58802 (Abcam) |
| CTSD | Rabbit monoclonal | 1:2000 | ab75852 (Abcam) |
| CTSG | Rabbit polyclonal | 1:100 | NBP2-33498 (Novus) |

imaged using an Olympus FV1200 biological confocal laser-scanning microscope (Olympus). All images were processed using cellSens 2.0 software (Olympus).

## Western blotting

Proteins were extracted from snap-frozen tissues and cell pellets, and western blotting (WB) was performed using Bolt 4–12% Bis-Tris gels (cat# NW04125BOX, ThermoFisher Scientific), an iBlot 2 apparatus (cat# IB21001, ThermoFisher Scientific) and an iBind apparatus (cat# SLF1000 or SLF2000, ThermoFisher Scientific), as described [27]. Primary antibodies used are outlined in Table 2. Secondary antibodies included: HRP-linked goat anti-rabbit (1:1000; cat# ab6721, Abcam), HRP-linked goat anti-rabbit (1:1000; cat# 111-035-045, Jackson Immunology), and Alexa Fluor® 488 donkey anti-mouse (1:1000; cat# A-21202, ThermoFisher Scientific).

Membranes were imaged using a ChemiDoc MP Imaging System (Biorad) and ImageLab 6.0 software (Biorad). Densitometry was performed using ImageLab 6.0, with the intensity values for the protein-of-interest normalized against α-tubulin. Densitometry data were analyzed using GraphPad Prism 8 (San Diego, CA, USA).

**Table 2. Primary antibodies used for western blotting.**

| Marker | Species/clonality | Dilution | Catalogue number |
|---|---|---|---|
| OCT4 | Rabbit monoclonal | 1:500 | ab109183 (Abcam) |
| SOX2 | Rabbit polyclonal | 1:1000 | 48–1400 (ThermoFisher Scientific) |
| NANOG | Rabbit monoclonal | 1:1000 | ab109250 (Abcam) |
| KLF4 | Rabbit polyclonal | 1:1000 | NBP2-24749 (Novus) |
| c-MYC | Rabbit monoclonal | 1:1000 | ab32072 (Abcam) |
| PRR | Rabbit polyclonal | 1:250 | ab40790 (Abcam) |
| ACE | Goat polyclonal | 1:200 | sc12184 (Santa Cruz) |
| ACE2 | Mouse monoclonal | 1:500 | MAB933 (R&D Systems) |
| AT$_2$R | Rabbit monoclonal | 1:500 | ab92445 (Abcam) |
| CTSB | Mouse monoclonal | 1:1000 | ab58802 (Abcam) |
| CTSD | Rabbit monoclonal | 1:1000 | ab75852 (Abcam) |
| α-tubulin | Mouse monoclonal | 1:2000 | ab7291 (Abcam) |

## Quantitative reverse transcription polymerase chain reaction

RNA was extracted using a QIAcube (Qiagen) according to the manufacturer's instructions using RLT lysis buffer and DTT from 20 mg of tissue or a pellet of $5x10^5$ cells. Following the run, the collection tube containing extracted RNA in 15 μL (cell pellets) of 45 μL (tissues) of buffer was retained. RNA was quantified using a NanoDrop 2000 spectrophotometer (ThermoFisher Scientific). A Rotor-Gene Q (Qiagen) was used for quantitative reverse polymerase chain reactions (RT-qPCR) according to the manufacturer's instructions, using 40 ng RNA per sample run in triplicate. The protocol ran as follows: reverse transcription at 50°C for 15 min; Taq polymerase activation at 95°C for 5 min; denature at 95°C for 15 sec and anneal and extend at either 60°C ($AT_2R$, PRR) or 62°C (OCT4, SOX2, NANOG, KLF4, c-MYC, ACE, CTSB, CTSD) for 15 sec (40 cycles). DMSO (5%) was added to the mastermix for OCT4, SOX2, NANOG, KLF4 and c-MYC, and 1 M Betaine was added to the mastermix for ACE, CTSB and CTSD.

## Cell culture

Primary cell lines derived from the LGCA and HGCA tissues used for IHC staining were provided by the GMRITB with approval by the Central Health and Disability Ethics Committee (Ref. 15/CEN/106). Cell lines were validated by comparison to their parent tissues via DNA sequencing [38]. CaCo2 (cat# HTB-37, ATCC, In Vitro Technologies, Auckland, New Zealand) cells were used as positive controls for tumorsphere formation assays. Cells were cultured in DMEM media with high glucose and containing pyruvate (cat# 10569010, ThermoFisher Scientific) and supplemented with 10% fetal calf serum (cat# 10091148, ThermoFisher Scientific), 5% mTeSR Complete (cat# 85850, STEMCELL Technologies, Tullamarine, Victoria, Australia), 1% penicillin-streptomycin (cat# 15140122, ThermoFisher Scientific) and 0.2% gentamicin/amphotericin B (cat# R01510, ThermoFisher Scientific).

**Cell sorting.** Cell lines were sorted into EpCAM$^{High}$ and EpCAM$^{Low}$ fractions using the CELLection™ Epithelial Enrich Dynabeads kit (cat# 16203, ThermoFisher Scientific), as described [27].

**Tumorsphere formation assays.** Tumorsphere formation assays were performed as described [27], in Corning Costar 6-well ultra-low attachment plates (cat# 3471, In Vitro Technologies) and T25 Nunclon Sphera EasyFlasks (cat# 174951, ThermoFisher Scientific) using StemXVivo Serum-free Tumoursphere media (cat# CCM012, R&D Systems, In Vitro Technologies) and seeded at a density of $1x10^4$ cells/mL.

**Cathepsin activity assays.** Cathepsin Activity Assay Kits (cat# ab65300 [CTSB] and cat# 65302 [CTSD], Abcam) were used according to the manufacturer's instructions with modifications outlined below. Total protein was extracted from tissues and cell samples using lysis buffers provided in each kit. The samples were washed briefly with cold PBS and homogenized in lysis buffer using an Axygen™ plastic pestle (cat# PSE-15-B-SI; ThermoFisher Scientific), then incubated shaking on ice for 20 min. For the CTSB assay, 100 μL of lysis buffer was added to 10 mg of tissue or $1x10^6$ cells. For the CTSD assay, 200 μL of lysis buffer was added to 100 mg of tissue or $1x10^6$ cells, and then 800 μL extra lysis buffer was added to the tissues after homogenization. Black-walled clear-bottom 96-well plates were set up with 50 μL of protein extract from each sample, in duplicate. Tonsil tissue was used as a positive control for both assays.

**RAS modulation assays.** RAS modulation assays were performed on 4 HGCA-derived primary cell lines using RealTime-Glo™ Cell Viability Assay (cat# Q9712, Promega, In Vitro Technologies) kits according to the manufacturer's instructions. The assay is non-lytic and lasts for 72 h, so the same plate could be read multiple times across72 h. Any increase or decrease in metabolism of <25% relative to the control cells (grown in the absence of any

drug) was considered to be within the natural variation expected between cultures and not a significant result.

Cells (1000/well) were seeded on day 1 in white-walled clear-bottom 96-well plates (cat# FAL353377, In Vitro Technologies), in triplicate for each dosage. Control wells were seeded in triplicate, and included cells grown in adjuvant + media and media alone to assess possible effects of the adjuvants on cell metabolism, as well as controls of adjuvant + media and media without cells to measure background luminescence. On day 2, NanoLuc® enzyme and substrate were added and the first luminescence reading was performed 1 h after the initial dose. On days 3 and 4, further doses of each drug were added to the cells and the plates were read again, at time points of 24 h and 48 h after the initial dose. The final reading was performed on day 5, 72 h after the initial dose, and the media was collected and stored.

To study the effects of RAS inhibition on tumorsphere formation and transcription of stemness-associated markers, 6000 cells per well were seeded in 24-well tissue culture plates. Doses were selected based on viability assay results to ensure that metabolism was affected but with minimal cell death. R-propranolol was dosed at 30 μM and 10 μM, R-timolol at 100 μM and 50 μM, EMA401 at 50 μM and 10 μM, and losartan and SMM02 at 100 μM. RNA was extracted from cells treated with all the above drug doses, whereas cells for tumorsphere assays were only treated with R-propranolol, EMA401 and losartan. Statistical analysis and graphing were performed on GraphPad Prism 8.

## Results

### ACE and ACE2 are variably expressed by CA tissues and CA-derived primary cell lines

ACE staining was weak in the CA epithelial cytoplasm, and more pronounced on the luminal surface of CA epithelial cell membranes (Fig 1A, 1C, 1E and 1G), but was not detected in NC tissues (Fig 1B, 1D, 1F and 1H) except for the endothelium of the microvessels. Overall, the staining was weaker in HGCA tissues (Fig 1E and 1G) compared to LGCA tissues (Fig 1A and 1C). In all 6 CA tissues, epithelial cells exhibited moderate cytoplasmic staining for ACE2 (Fig 1I, 1K, 1M and 1O), with stronger membranous staining of the luminal surface and occasionally the stromal surface. Only 1 NC sample displayed positive staining (Fig 1N), on the luminal surface of crypt epithelial cells.

ACE was detected by RT-qPCR in all LGCA and HGCA tissues and patient-matched NC tissues with variable relative abundances (Fig 1Q). In the cell lines, ACE mRNA was detected by RT-qPCR in all CA-derived EpCAM$^{High}$ (+) and EpCAM$^{Low}$ (-) cells, but at significantly lower levels than the pooled NC tissues used as a reference (Fig 1R). ACE2 mRNA was detected in all cell lines except LGCA3 EpCAM$^{Low}$ (-) cells and LGCA1 EpCAM$^{High}$ (+) cells. However, the abundance was much lower than in the pooled NC tissue reference (Fig 1S).

ACE and ACE2 were detected by WB in the positive controls at approximately 195 kDa and 110 kDa, respectively (Fig 1T and 1U), but not in CA-derived cell lines.

### AT$_2$R expression in CA tissues increases with tumor grade

Staining for AT$_2$R across the CA samples (Fig 2A, 2C, 2E and 2G) ranged from negative to strong, with most positive cases displaying uniform moderate cytoplasmic staining of the epithelial cells, and others displaying a more granular cytoplasmic staining pattern. In NC tissues (Fig 2B, 2D, 2F and 2H), staining was overall negative to weak, and seemed to be cytoplasmic or concentrated to the luminal membrane of epithelial cells when present, with some nuclear

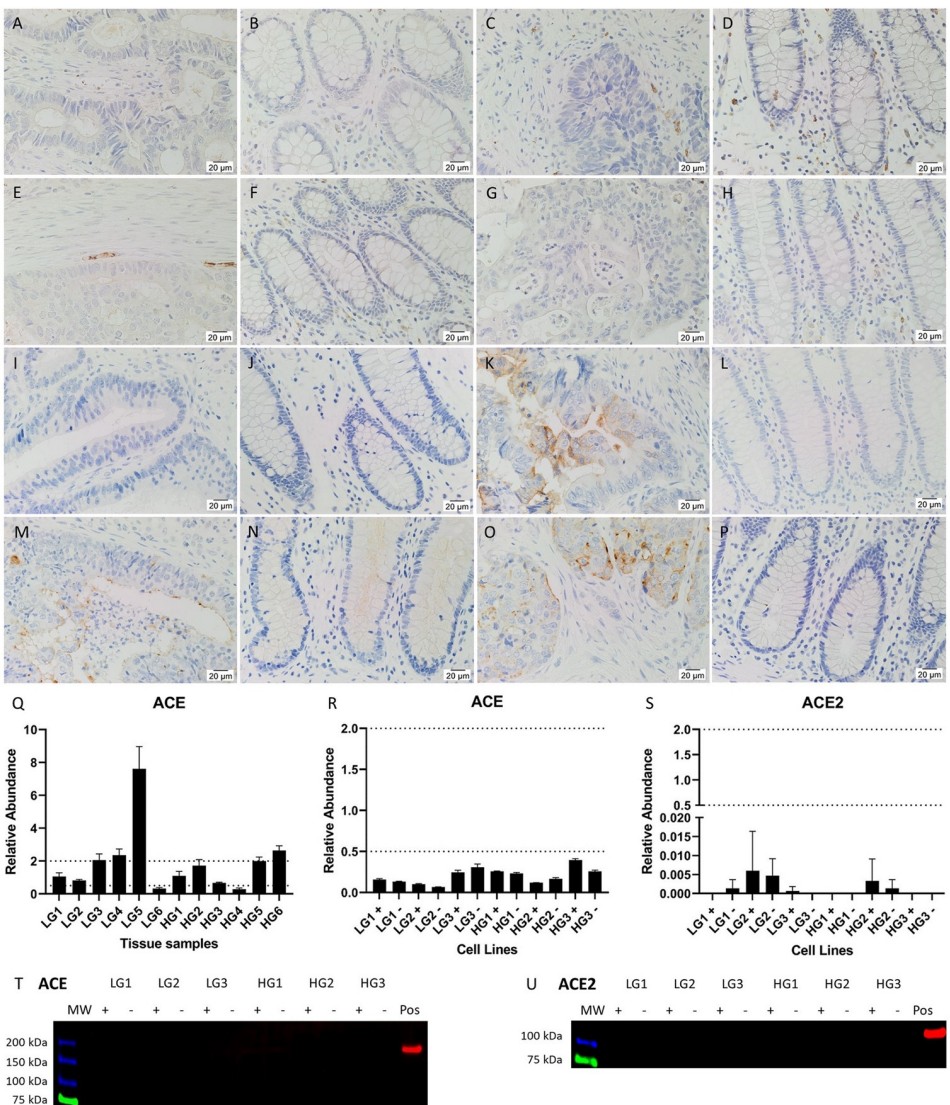

**Fig 1. Expression of ACE and ACE2.** Representative IHC stained images of ACE in 2 LGCA samples (A, C) and 2 HGCA samples (E, G) and their patient-matched NC) tissues (B, D, F, H), and ACE2 in 2 LGCA samples (I, K) and 2 HGCA samples (M, O) and their patient-matched NC) tissues (J, L, N, P). Nuclei were counterstained with hematoxylin (A-H, blue); original magnification: 400x. ACE (R) and ACE2 (S) mRNA expression levels were measured by RT-qPCR in 3 LGCA and 3 HGCA primary cell lines that were sorted into EpCAM$^{High}$ (+) and EpCAM$^{Low}$ (-) subpopulations. ACE mRNA expression levels were also measured in 6 LGCA and 6 HGCA tissues (Q). Abundance in CA tissues is displayed relative to patient-matched NC tissues, and relative to a pool of NC tissues for the CA-derived cell lines, with error bars representing standard deviation. ACE (T; 195 kDa) and ACE2 (U; 110 kDa) protein expression by cell lines could not be detected by WB; positive controls = mouse lung and human kidney, respectively. Adapted from [38] under a CC BY license, with permission from Munro MJ *et al.* 2021.

staining. Some stromal cells stained positively, most of which are immune cells, although others were elongated fibroblast-like cells.

AT$_2$R mRNA was detected in 2 of the 6 LGCA tissues, with variable relative abundance (Fig 2I). It was detected in 5 of the 6 HGCA tissues but only in 2 of their patient-matched NC samples. AT$_2$R mRNA was below detectable levels in all CA-derived EpCAM$^{High}$ (+) and EpCAM$^{Low}$ (-) cell lines (Fig 2J).

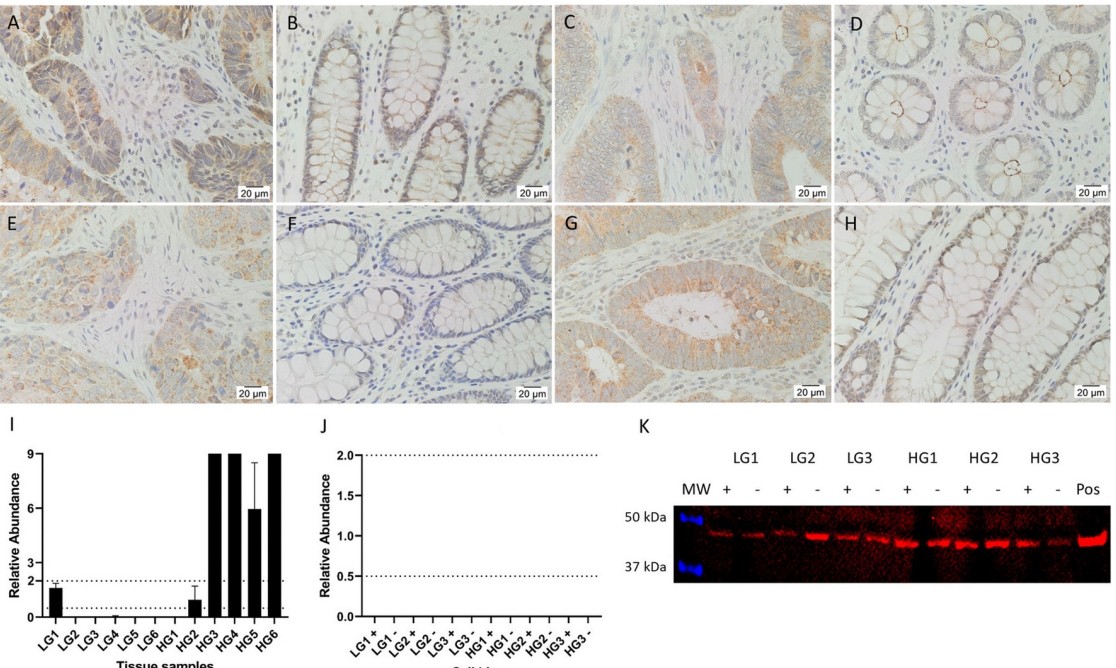

**Fig 2. Expression of AT$_2$R.** Representative IHC stained images of AT$_2$R (brown), including 2 LGCA samples (A, C) and their patient-matched NC tissues (B, D), and 2 HGCA samples (E, G) and their patient-matched NC tissues (F, H). Nuclei were counterstained with hematoxylin (A-H, blue); original magnification: 400x. AT$_2$R mRNA expression levels were measured by RT-qPCR in 6 LGCA and 6 HGCA tissues (I) and in 3 LGCA and 3 HGCA primary cell lines that were sorted into EpCAM$^{High}$ (+) and EpCAM$^{Low}$ (-) subpopulations (J). Abundance in CA tissues is displayed relative to patient-matched NC tissues, and relative to a pool of NC tissues for the CA-derived cell lines, with error bars representing standard deviation. PRR protein expression by cell lines was detected by WB (K; 45 kDa); positive control = HepG2 cells. Adapted from [38]. under a CC BY license, with permission from Munro MJ *et al.* 2021.

A band corresponding to AT$_2$R was detected by WB in all EpCAM$^{High}$ and EpCAM$^{Low}$ cells (Fig 2K) at approximately 45 kDa.

## PRR is upregulated in CA tissues and CA-derived cell lines

PRR staining in CA tissues (Fig 3A, 3C, 3E and 3G) was predominantly moderate in the cytoplasm of epithelial cells, and often moderate to strong on the cell membrane. NC tissues (Fig 3B, 3D, 3F and 3H) stained strongly in the muscularis mucosae below the crypts and stroma (Fig 3B), and negative or weak in the cytoplasm of crypt epithelial cells.

PRR mRNA was detected in NC, LGCA and HGCA tissues with variable relative abundance (Fig 3I). All CA-derived EpCAM$^{High}$ (+) and EpCAM$^{Low}$ (-) cells expressed significantly more PRR mRNA than the pooled NC tissue reference (Fig 3J).

PRR was detected by WB at the expected molecular weight of 35 kDa (Fig 3K) in all EpCAM$^{High}$ and EpCAM$^{Low}$ cell lines.

## Cathepsins are expressed in the CA epithelium and stroma

IHC staining showed that CTSB (Fig 4A–4H) and CTSD (Fig 5A–5H) were highly expressed by immune cells in the stroma of CA and patient-matched NC tissues, and by occasional crypt cells, presumably neuroendocrine cells, in NC tissues. CA epithelial cells showed weak to moderate staining of both CTSB (Fig 4A, 4C, 4E and 4G) and CTSD (Fig 5A, 5C, 5E and 5G), with CTSD appearing more granular with spots of strong staining, possibly lysosomes or

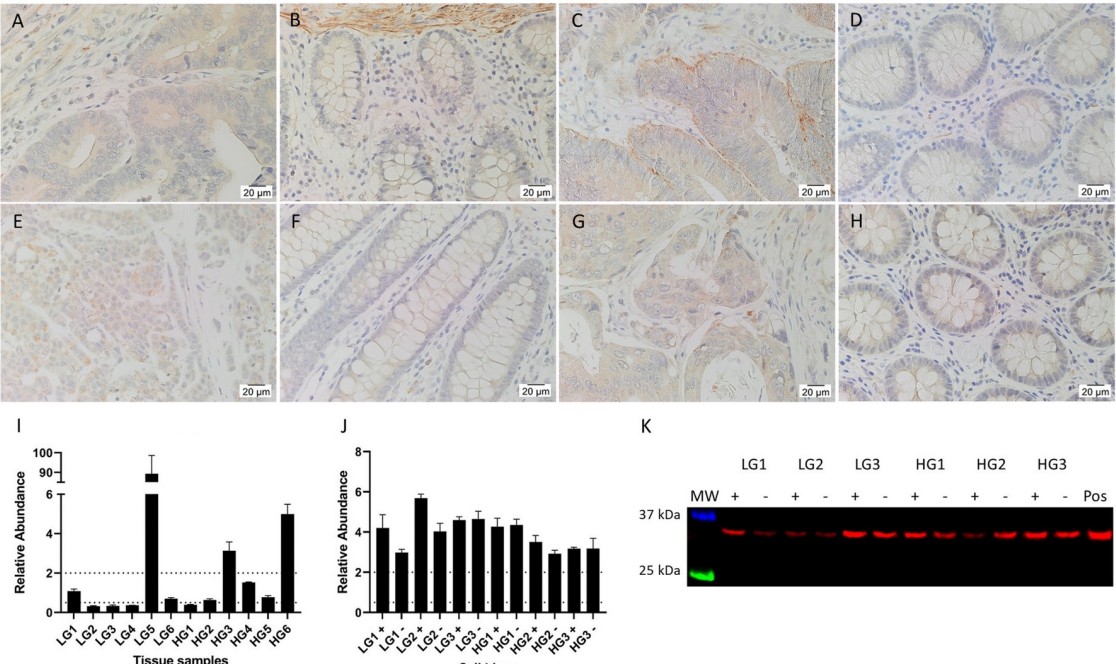

**Fig 3. Expression of PRR.** Representative IHC stained images of PRR (brown), including 2 LGCA samples (A, C) and their patient-matched NC tissues (B, D), and 2 HGCA samples (E, G) and their patient-matched NC tissues (F, H). Nuclei were counterstained with hematoxylin (A-H, blue); original magnification: 400x. PRR mRNA expression levels were measured by RT-qPCR in 6 LGCA and 6 HGCA tissues (I) and in 3 LGCA and 3 HGCA primary cell lines that were sorted into EpCAM$^{High}$ (+) and EpCAM$^{Low}$ (-) subpopulations (J). Abundance in CA tissues is displayed relative to patient-matched NC tissues, and relative to a pool of NC tissues for the CA-derived cell lines, with error bars representing standard deviation. PRR protein expression by cell lines was detected by WB (K; 35 kDa); positive control = tonsil. Adapted from [38] under a CC BY license, with permission from Munro MJ *et al*. 2021.

endosomes in which cathepsins generally function. Staining of both CTSB and CTSD was stronger in the epithelial cells of LGCA tissues than HGCA tissues. NC epithelial cells expressed low levels of CTSB (Fig 4B, 4D, 4F and 4H). CSTD was weakly positive in the epithelium of a minority of NC tissues (Fig 5B, 5D, 5F and 5H). CTSG (Fig 6) was expressed by cells scattered within the stroma, thought to be mast cells which typically express CTSG [39–44]. The stain also appeared non-specifically in necrotic areas (Fig 6A).

RT-qPCR detected mRNA for CTSB and CTSD in all CA-derived EpCAM$^{High}$ (+) and EpCAM$^{Low}$ (-) cells. All cell lines had significantly higher levels of CTSB mRNA relative to the pooled NC tissue reference (Fig 4I). CTSD expression across the cell lines was equal to or greater than that of the pooled NC (Fig 5I).

Bands for CTSB (Fig 4J) were detected by WB in all CA-derived EpCAM$^{High}$ and EpCAM$^{Low}$ cell lines. Two prominent bands were observed, corresponding to the heavy chain of CTSB with and without glycosylation (27 kDa and 24 kDa, respectively). CTSD (Fig 5J) was detected in all HGCA-derived cell lines, and in two of the three LGCA-derived cell lines (both EpCAM$^{High}$ and EpCAM$^{Low}$ cells), where it was predominantly more abundant in EpCAM$^{High}$ cells. Bands of two different sizes were detected, which represented pre-pro-cathepsin D (43 kDa), and glycosylated pro-cathepsin D (46 kDa).

## OCT4$^+$ and NANOG$^+$ cells express RAS components

IF staining was carried out using combinations of various stemness-associated markers with RAS components to determine whether CA stem-like cells express RAS components.

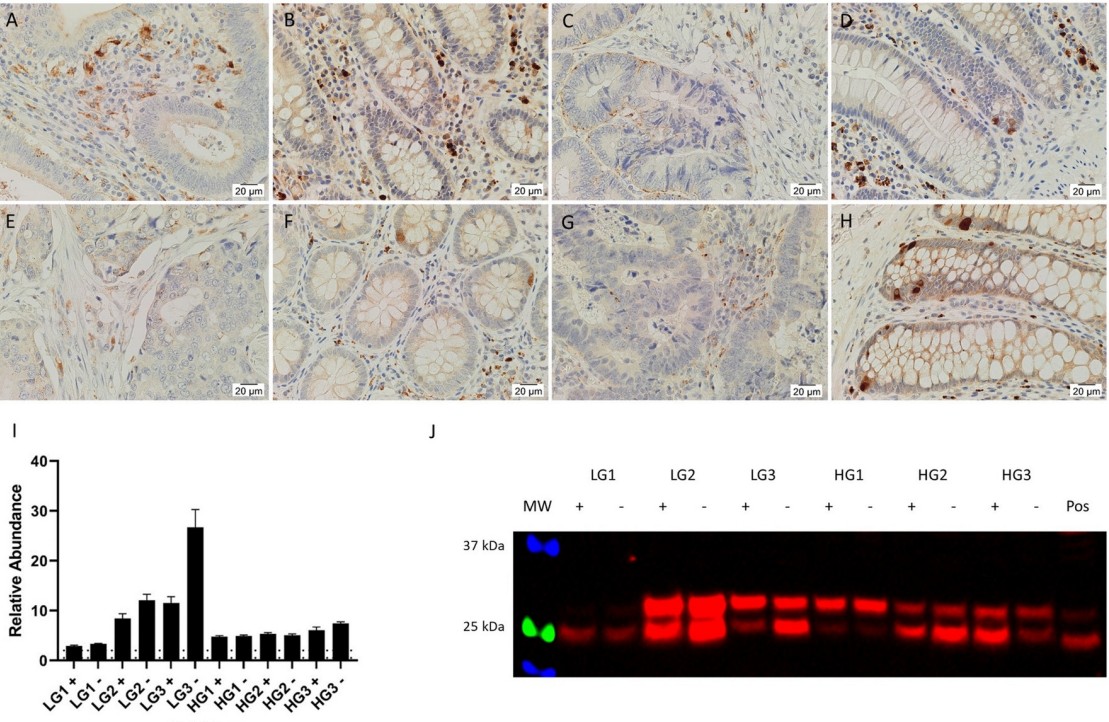

**Fig 4. Expression of cathepsin B.** Representative IHC stained images of CTSB, including 2 LGCA samples (A, C) and their patient-match NC tissues (B, D), and 2 HGCA samples (E, G) and their patient-matched NC tissues (F, H). Nuclei were counterstained with hematoxylin (A-H, blue); original magnification: 400x. CTSB mRNA expression levels were measured by RT-qPCR (I) in 3 LGCA and 3 HGCA primary cell lines that were sorted into EpCAM$^{High}$ (+) and EpCAM$^{Low}$ (-) subpopulations, and the average mRNA abundance from triplicate values relative to a pool of NC tissues are displayed with error bars representing standard deviation. CTSB protein expression by cell lines was detected by WB (J; 24 kDa and 27 kDa); positive control = HepG2 cells. Adapted from [38] under a CC BY license, with permission from Munro MJ *et al.* 2021.

OCT4 (Fig 7A–7I; green) was expressed by neuroendocrine cells within the crypts of NC tissues [45], and in the cytoplasm of elongated stromal cells in LGCA and HGCA tissues. Some of these OCT4$^+$ cells showed nuclear staining of AT$_2$R (Fig 7B and 7C; red; *arrowheads*). AT$_2$R was also present in the cytoplasm and plasma membrane of epithelial cells in CA tissues.

CTSB (Fig 7D–7F; red) and CTSD (Fig 7G–7I; red) were expressed in the cytoplasm of epithelial cells in CA and NC tissues, and abundantly in stromal immune cells. There did not appear to be any co-expression of either CTSB or CTSD with OCT4 in the CA stromal cell subpopulation.

NANOG (Fig 7J–7L; red) was seen in the cytoplasm and nuclei of epithelial cells in CA tissues but not NC tissues. ACE2 (Fig 7J–7L; green) was localized to the cytoplasm and luminal membrane of CA crypt epithelial cells that also expressed NANOG. The mouse anti-NANOG primary antibody (Fig 7M–7O; green) produced a weaker stain than the rabbit anti-NANOG (Fig 7J–7L; red), but was detected in the cytoplasm of tumor epithelial cells, which showed cytoplasmic and membranous expression of AT$_2$R (Fig 7M–7O; red).

Overall, IF staining showed that the stromal OCT4$^+$ and epithelial NANOG$^+$ subpopulations [28] both express AT$_2$R, and that the NANOG$^+$ subpopulation also expressed ACE2, CTSB and CTSD. Based on the results of this study and our previous data [28], the proposed epithelial CSC-like subpopulation expresses NANOG, SOX2, KLF4, c-MYC, CD133, EpCAM, AT$_2$R and ACE2, and possibly LGR5, PRR, CTSB and CTSD.

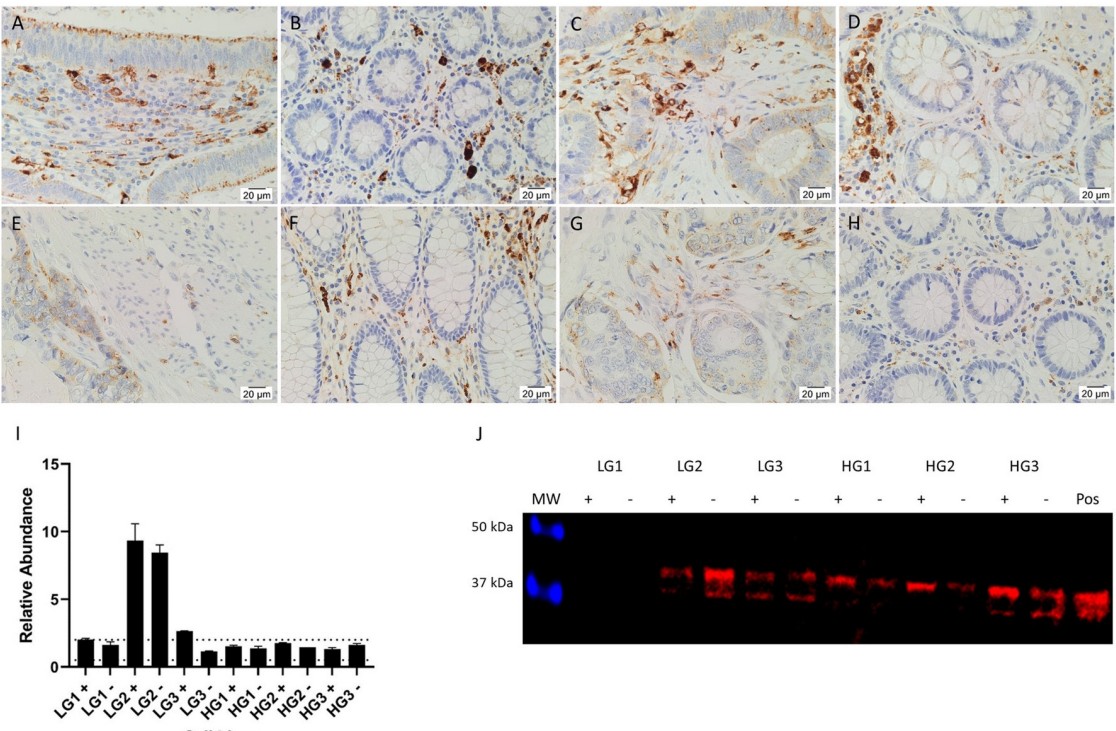

**Fig 5. Expression of cathepsin D.** Representative IHC stained images of CTSD (brown), including 2 LGCA) samples (A, C) and their patient-match NC tissues (B, D), and 2 HGCA samples (E, G) and their patient-matched NC tissues (F, H). Nuclei were counterstained with hematoxylin (A-H, blue); original magnification: 400x. CTSD mRNA expression levels were measured by RT-qPCR (I) in 3 LGCA and 3 HGCA primary cell lines that were sorted into EpCAM$^{High}$ (+) and EpCAM$^{Low}$ (-) subpopulations, and the average mRNA abundance from triplicate values relative to a pool of NC tissues are displayed with error bars representing standard deviation. CTSD protein expression by these cell lines was detected by WB (J; 43 kDa and 46 kDa); positive control = HepG2 cells. Adapted from [38] under a CC BY license, with permission from Munro MJ *et al.* 2021.

## Cathepsins B and D in CA tissues and CA-derived primary cell lines are active

Activity of CTSB and CTSD was detected in all CA tissues and CA-derived primary cell lines (S2A–S2D Fig). In CA tissues, CTSB activity was 2.84–6.95 FIU/µg of protein, with tonsil having 3.58 FIU/µg (S2A Fig). CTSD activity was 0.09–0.56 FIU/µg, with tonsil having 0.41 FIU/µg (S2B Fig). It was expected that the tissues would have higher cathepsin activity due to the presence of immune cells throughout the stroma as identified by IHC staining. However, the cells had higher cathepsin activity per µg of total protein than the tissues. The range of activity in the cells was 2.27–33.23 FIU/µg for CTSB (S2C Fig) and 0.61–2.70 FIU/µg for CTSD (S2D Fig). These results showed that CTSB and CTSD were active in CA tissues and CA-derived primary cell lines and therefore may be capable of functioning in the RAS.

## RAS inhibitors affect cancer cell metabolism

Propranolol administered at 50 µM consistently led to a 96–99% reduction in cell metabolism and visible cell death in the HGCA-derived primary cells (Fig 8A–8D and S3A–S3L Fig), an effect greater than expected given the IC50 of approximately 65 µM reported previously [46]. In contrast, timolol, which is reportedly 10x as potent as propranolol at antagonizing β-adrenergic receptors [47], had a lesser effect on cell metabolism, with only the 100 µM doses causing a decrease of >25% (Fig 8E–8H and S4A–S4H Fig). The two enantiomers of each β-blocker

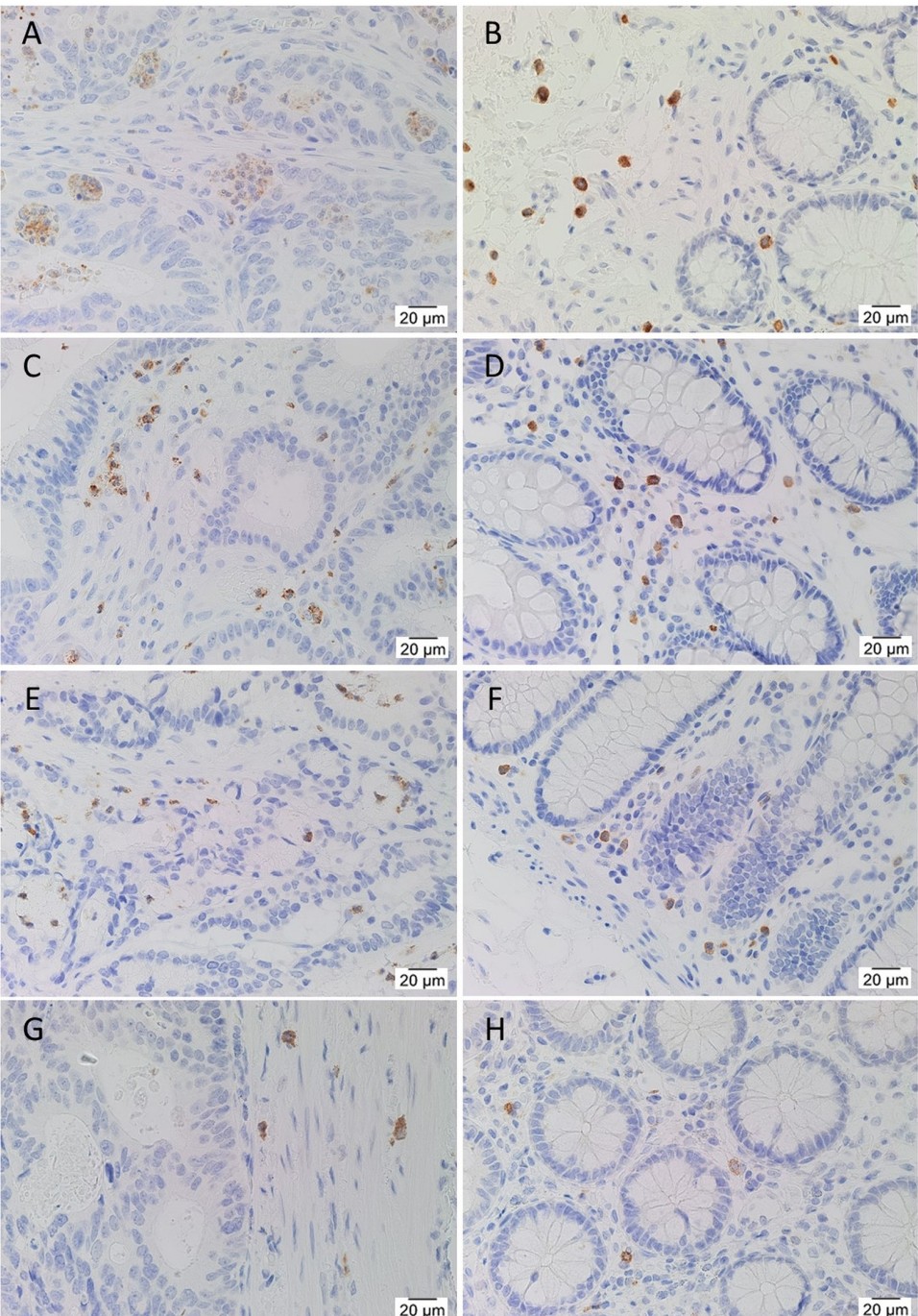

**Fig 6. IHC staining for cathepsin G.** Representative IHC stained images of CTSG (brown), including 2 LGCA samples (A, C) and their patient-match NC tissues (B, D), and 2 HGCA samples (E, G) and their patient-matched NC tissues (F, H). Nuclei were counterstained with hematoxylin (A-H, blue); original magnification: 400x. Reproduced from [38] under a CC BY license, with permission from Munro MJ *et al*. 2021.

produced almost identical results. The R- and S-enantiomers of propranolol were administered separately and as a racemic mixture, and a 50 µM dose of either enantiomer alone or of the racemic combination caused >90% inhibition of metabolic activity by 48 h in all 4 cell lines. Doses of 10 µM and 1 µM did not make a significant difference at any time point.

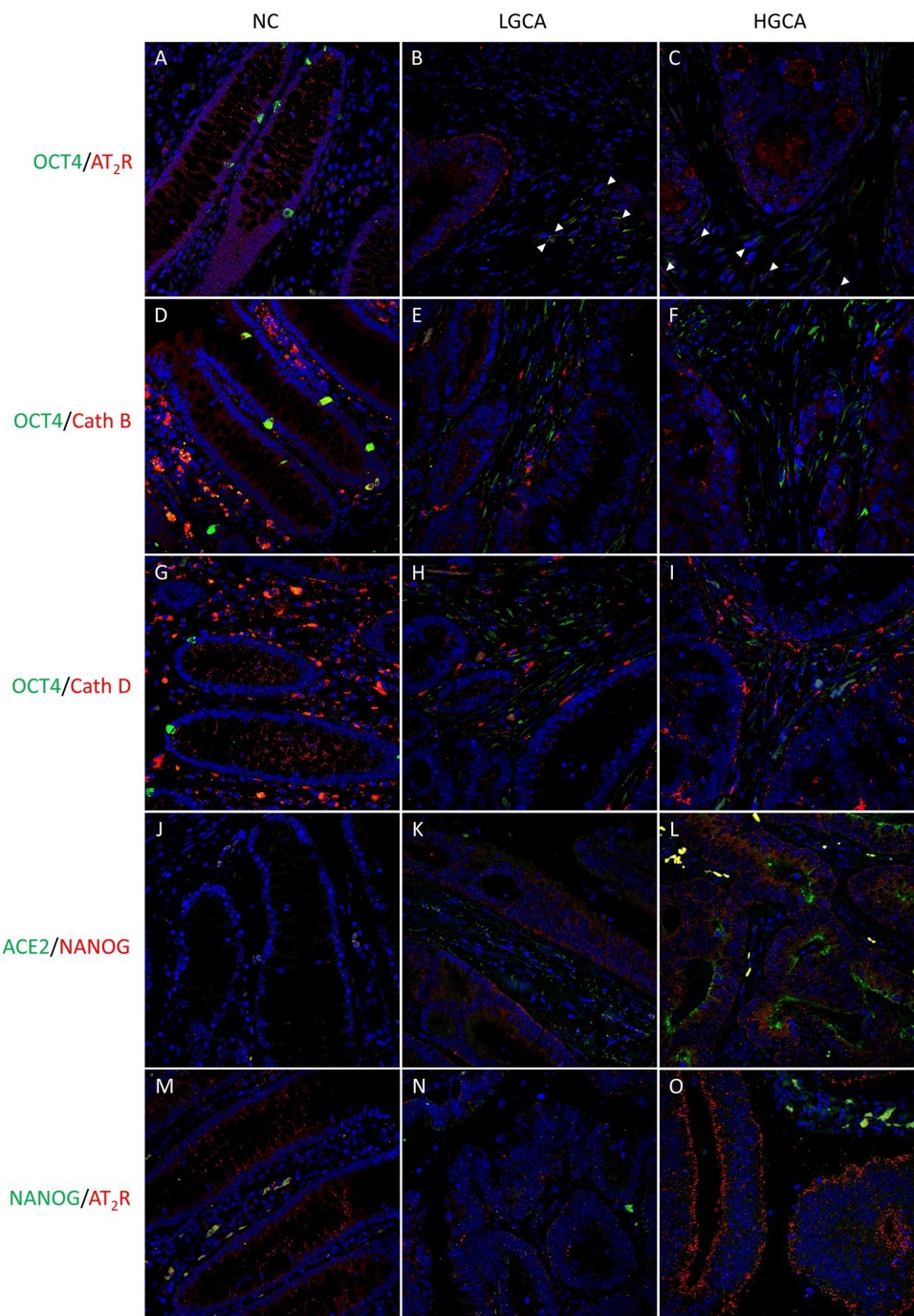

**Fig 7. Co-localization of stemness-associated markers with renin-angiotensin system components by IF staining.**
Representative IF images showing protein expression of OCT4 (A-I, green) with AT$_2$R (A-C, red), cathepsin B (D-F, red) and cathepsin D (G-I, red). A rabbit anti-NANOG antibody (J-L, red) was co-stained with ACE2 (J-L, green). A mouse anti-NANOG antibody (M-O, green) was co-stained with AT$_2$R (A-C, red). Normal colon (NC; A,D,G,J,M), low-grade colon adenocarcinoma (LGCA; B,E,H,K,N), high-grade colon adenocarcinoma (HGCA; C,F,I,L,O). Cell nuclei were counterstained with 4′,6-diamidino-2-phenylindole (A-O, blue). Original magnification: 400x. Reproduced from [38] under a CC BY license, with permission from Munro MJ *et al.* 2021.

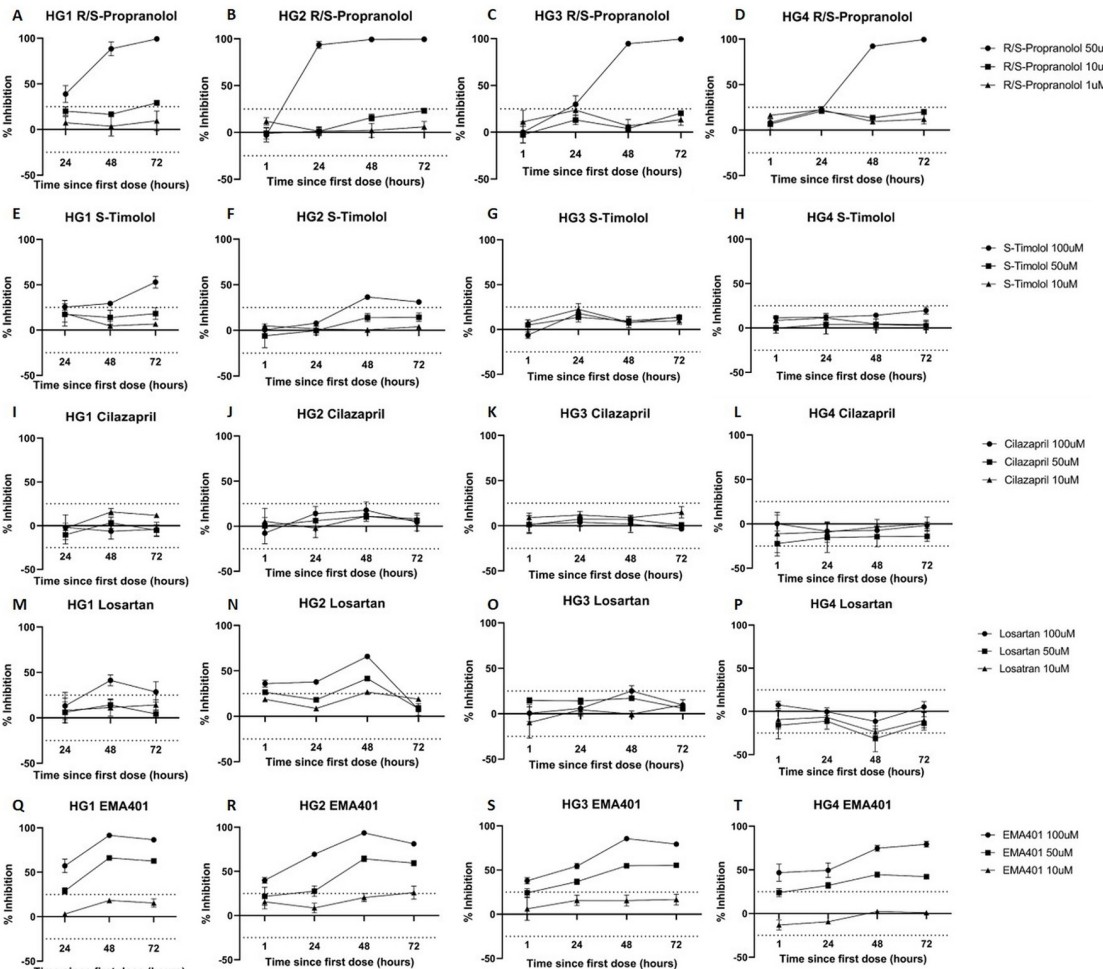

**Fig 8. Effect of RASIs on the metabolism of HGCA-derived primary cell lines.** Four HGCA-derived cells lines were exposed to racemic propranolol (A-D), S-timolol (E-H), cilazapril (I-L), losartan (M-P) and EMA401 (Q-T) at doses of 50 μM (●), 10 μM (■) and 1 μM (▲). Luminescence produced by cells exposed to RASIs was detected and used to measure the inhibitory effect of treatment on metabolic activity. The y-axis represents the extent of metabolic inhibition in the treated cells relative to the untreated control cells, and because the assay seeks to measure metabolic inhibition, the positive values indicate inhibition and the negative values indicate increased metabolic activity in the drug-treated cells. The x-axis shows the time after initial dose of each luminescence measurement. Each data point is the average of 3 technical replicates, with error bars showing standard deviation. Adapted from [38] under a CC BY license, with permission from Munro MJ *et al*. 2021.

Similarly, 72 h after a 100 μM dose, metabolism was inhibited by 30–50% by R-timolol in 3 cell lines, and by S-timolol in 2 of these same cell lines. The metabolism of the HGCA3-derived cell line was unaffected by any dose of either timolol enantiomer (S4C and S4G Fig).

ACEIs were administered to the cell lines due to reports that they reduce the incidence and mortality of CRC. However, due to the ambiguity around ACE protein expression, ACEIs were not expected to affect the metabolism of the CA-derived primary cell lines. Unsurprisingly, neither captopril nor cilazapril caused any significant changes to metabolism in any of the 4 CA cell lines at any concentration (Fig 8I–8L and S5A–S5H Fig).

There are currently no reliable antibodies for $AT_1R$ [48] and so its presence in these samples is undetermined. However, previous work suggests that ARBs are beneficial to CRC patients. Surprisingly, candesartan (S6A–S6D Fig) and losartan (Fig 8M–8P) did not affect cell metabolism in any of the 4 cell lines by the 72 h time point. The HGCA1 and HGCA2 cell lines

appeared to have lower metabolism relative to control cells 48 h after beginning daily 100 μM doses of either candesartan or losartan, but this had normalized by 72 h, suggesting that the cells may have been able to overcome any early effects of the drugs despite daily dosing (S6A, S6B, S6E and S6F Fig).

EMA401 led to a clear reduction of metabolism, with the level of inhibition proportional to the dosage administered (Fig 8Q–8T). In all 4 cell lines, doses of 50 μM and 100 μM consistently caused reductions in metabolism of around 80% and 50%, respectively, whereas the 10 μM dose did not affect metabolism. SMM02 was trending in the same direction until the 72 h time point, where despite the inhibition remaining above 25% for the 100 μM dose in all 4 cell lines, metabolism seemed to be recovering (S7E–S7H Fig).

Overall, R-, S- and racemic propranolol at doses of 50 μM, and EMA401 at doses of 50 μM and 100 μM, consistently reduced cellular metabolism. ACEIs did not affect metabolism, and ARBs had an initial influence in 2 CA-derived primary cell lines before metabolism returned to levels similar to control cells. SMM02 might also inhibit metabolism, but the cells appeared to be adjusting or recovering partially by 72 h.

## RAS inhibitors reduce tumorsphere formation

HGCA-derived primary cell lines were treated with R-propranolol (20 μM), EMA401 (50 μM) and losartan (100 μM) at concentrations estimated to cause a significant change in metabolism without extensive cell death, to allow a sufficient number of cells to be harvested and seeded for tumorsphere assays.

An average diameter of 50 μm was chosen as the threshold for successful tumorsphere formation by a cell line [27]. Untreated control cells from all 4 cell lines were able to reach the threshold in the first passage (Fig 9A–9D), and while all 4 cell lines formed sphere-like structures in the second passage, only HGCA3 reached the 50 μm threshold (Table 3). When exposed to losartan, each cell line still formed tumorspheres with average diameters near or greater than the threshold (Fig 9E–9H), but could not be recapitulated in a second passage (Table 3). HGCA1 cells treated with losartan produced larger tumorspheres than the control cells in passage 1 but failed in passage 2. Cells exposed to EMA401 reached the threshold in HGCA1 but fell just short in the other 3 cell lines (Fig 9I–9L and Table 3). Similarly, cells treated with R-propranolol reached the threshold in 2 cell lines (Fig 9M–9P and Table 3). Overall, the RAS modulators had a variable effect on tumorsphere formation in the first passage, but appeared to limit the ability of cells to recapitulate tumorspheres in the second passage, with EMA401 reducing tumorsphere size most consistently.

## RAS inhibitors reduce expression of pluripotency genes

Cells were grown in 24-well plates and exposed to R-propranolol (30 μM and 10 μM), R-timolol (100 μM and 50 μM), EMA401 (50 μM and 10 μM), SMM02 (100 μM) and losartan (100 μM), before lysis and RNA collection.

*OCT4* mRNA expression (Fig 10A) was lower in the untreated control cells than the pooled NC tissues, and was not detected in HGCA3 untreated cells. However, cells exposed to all 5 RAS modulators expressed significantly less OCT4 than untreated cells in 3 of the cell lines. Conversely, OCT4 mRNA was detected in treated HGCA3 cells despite being below the detection threshold in untreated cells.

*SOX2* mRNA (Fig 10B) was expressed in 3 of the untreated cell lines, but was below detectable levels in HGCA3. Exposure to any of the RAS modulators caused a reduction in SOX2 mRNA levels, which could not be detected in any treated cells.

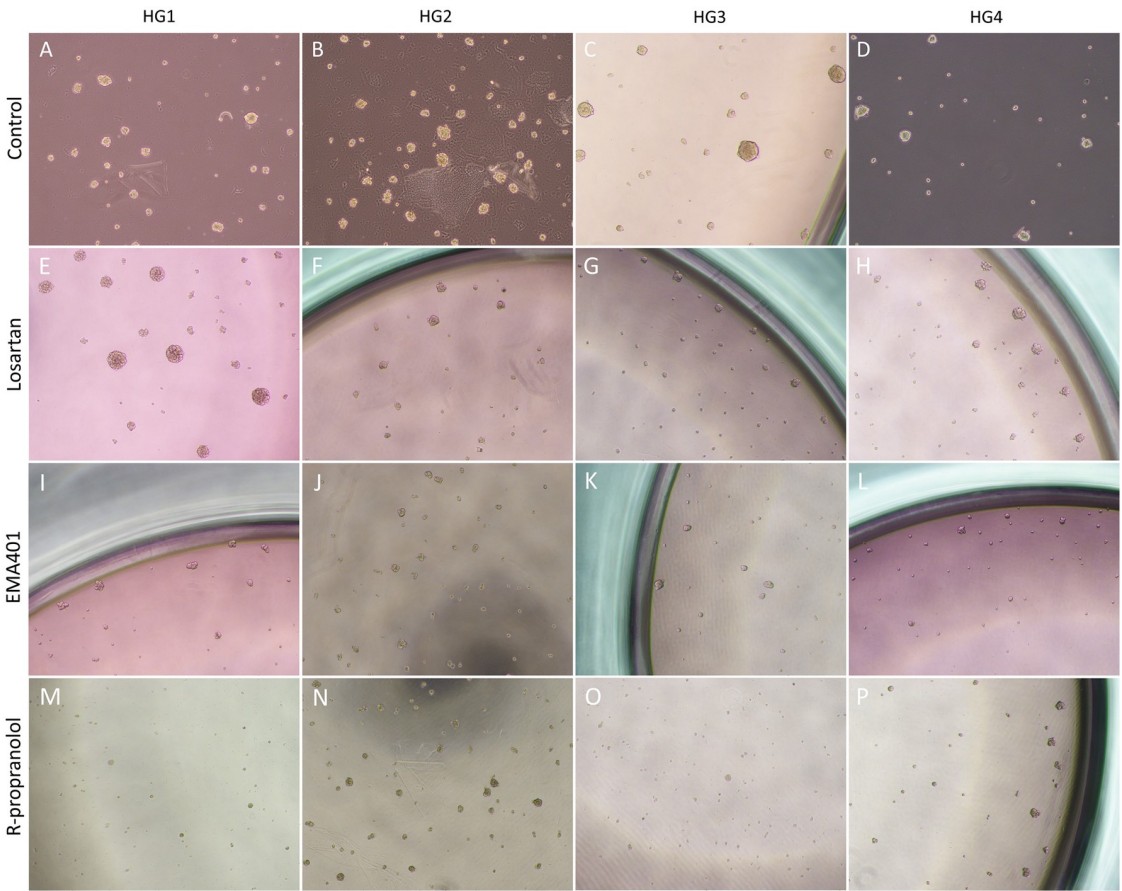

**Fig 9. Effect of RASIs on tumorsphere formation.** Four HGCA-derived primary cell lines were exposed to losartan (E-H), EMA401 (I-L) and R-propranolol (M-P). Untreated cells were used as a control (A-D). Images show first-passage tumorspheres. Original magnification: 100x; scale bar: 200 μm. Reproduced from [38] under a CC BY license, with permission from Munro MJ *et al*. 2021.

Similarly, *NANOG* (Fig 10C) was present in all 4 untreated cell lines, though at very low levels in HGCA3, and was below the detection threshold in all cells treated with any of the RAS modulators tested.

KLF4 expression levels are reportedly highest in the NC, with decreased levels in LGCA and the lowest expression in HGCA [49]. Cells treated with 50 μM doses of EMA401 or 100 μM doses of SMM02 exhibited upregulation of *KLF4* mRNA in 4 and 3 cell lines, respectively (Fig 10D). Similarly, 30 μM R-propranolol or 100 μM R-timolol caused an increase in *KLF4* mRNA levels in 3 and 4 cell lines, respectively. All 5 RAS modulators at all concentrations led to increased *KLF4* mRNA levels in the HGCA1 and HGCA2 cell lines.

Overall, the RAS modulators caused a reduction in the mRNA levels of pluripotency markers *OCT4*, *SOX2* and *NANOG*, and increased the expression of the goblet lineage differentiation-associated marker *KLF4*. Despite the lack of effect of the ARBs on metabolism, losartan slightly hindered tumorsphere formation in 3 cell lines and changed the expression patterns of stemness-associated markers. R-propranolol, EMA401 and SMM02 inhibited metabolism and altered stemness-associated marker transcription, while R-propranolol and EMA401 also reduced the size of tumorspheres formed by treated cells relative to untreated controls.

**Table 3. Analysis of tumorsphere formation from treated cells.**

| Sample | Passage | Average maximum diameter ± SD | | | |
|---|---|---|---|---|---|
| | | Control | R-propranolol | EMA401 | Losartan |
| HGCA1 | 1 | 55.39 (3) ± 11.92 | 29.06 (4) ± 4.76 | 50.71 (4) ± 15.95 | 74.24 (4) ± 26.07 |
| | 2 | 32.30 (2) ±7.44 | N/A | 34.22 (2) ± 4.43 | N/A |
| | 3 | N/A | N/A | N/A | N/A |
| HGCA2 | 1 | 60.38 (7) ± 12.86 | 50.58 (3) ± 6.80 | 45.15 (3) ± 9.02 | 51.17 (3) ± 11.78 |
| | 2 | 44.32 (3) ± 9.77 | N/A | N/A | N/A |
| | 3 | N/A | N/A | N/A | N/A |
| HGCA3 | 1 | 72.20 (4) ± 18.87 | 42.75 (3) ± 5.10 | 43.87 (4) ± 10.70 | 46.06 (3) ± 9.72 |
| | 2 | 53.92 (4) ± 13.70 | N/A | N/A | N/A |
| | 3 | 32.25 (3) ± 7.56 | N/A | N/A | N/A |
| HGCA4 | 1 | 59.52 (4) ± 14.21 | 52.90 (3) ± 11.15 | 41.31 (3) ± 11.12 | 56.82 (3) ± 12.32 |
| | 2 | 38.12 (6) ± 7.29 | 31.75 (3) ± 3.70 | N/A | N/A |
| | 3 | N/A | N/A | N/A | N/A |

Primary cell lines derived from 4 HGCA tissue samples were exposed to 3 RAS modulators before being seeded for tumorsphere assays. Tumorsphere diameter was measured in μm. The number of days taken to reach the maximum diameter are shown in brackets. Diameter values represent the average diameter of all measured tumorspheres across multiple technical replicates for each biological replicate, with SD values shown.

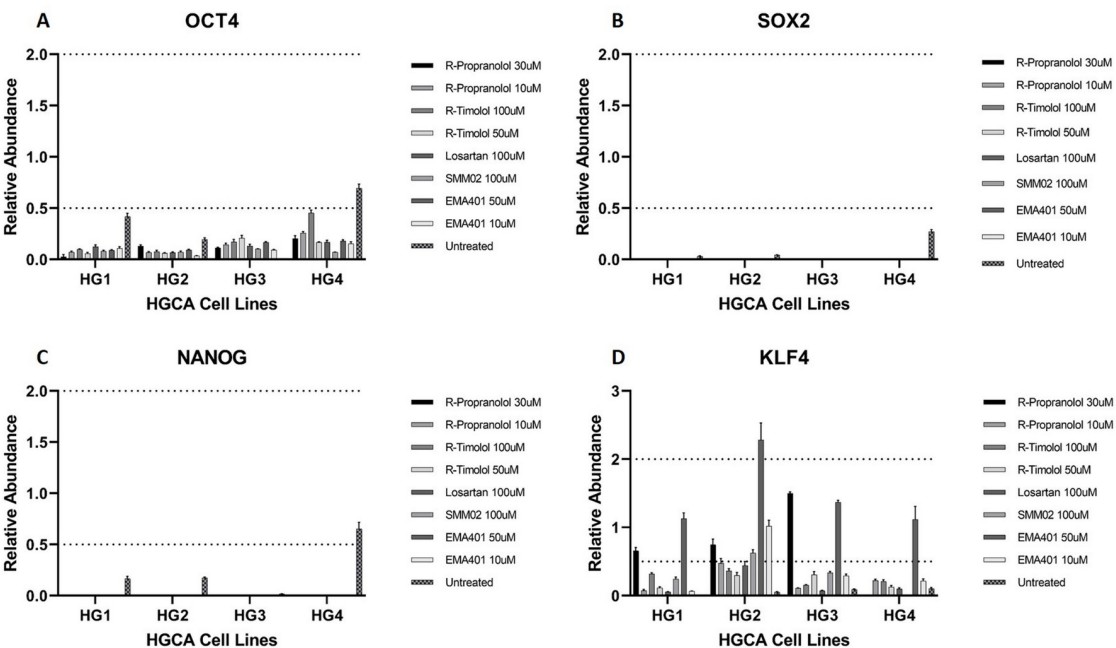

**Fig 10. mRNA levels of stemness-associated markers in HGCA-derived cell lines treated with RASIs.** RNA was extracted from 4 high-grade colon adenocarcinoma (HGCA)-derived primary cell lines that had been exposed to RASIs, as well as untreated controls, and the abundance of *OCT4* (A), *SOX2* (B), *NANOG* (C) and *KLF4* (D) mRNA from these cell lines relative to the normal colon (NC) tissues was calculated (y-axis). Error bars show standard deviation from the average of technical replicates after removing outliers (CT value +/- 0.5 from average). Reproduced from [38] under a CC BY license, with permission from Munro MJ *et al*. 2021.

## Discussion

This study investigated the expression and localization of RAS components and whether there were any differences between CA tissues and their patient-matched NC tissues. Dual IF

staining of stemness-associated markers with RAS components was performed to investigate if these components were expressed by the NANOG$^+$ CA CSC-like subpopulation on the tumor epithelium and by an OCT4$^+$ cell subpopulation in the stroma of CA tissues. We then assessed the effects of RAS modulators on the metabolism, tumorsphere forming capacity and transcription of stemness-associated markers in HGCA-derived primary cell lines. We showed that β-blockers, but not ACEIs, reduce cell metabolism. Unexpectedly, AT$_2$R antagonists, but not ARBs, caused a reduction in metabolic activity. Losartan, along with R-propranolol, R-timolol, EMA401 and SMM02, reduced transcription of stemness-associated markers.

IHC staining revealed higher abundance of PRR, CTSB and CTSD in CA tissues than their patient-matched NC samples, predominantly in the cytoplasm or luminal membrane of epithelial cells. Unexpectedly, ACE2 and AT$_2$R proteins were also more abundant in CA tissues. Interestingly, CTSB and CTSD were more abundant in the epithelium of LGCA than HGCA tissues, suggesting that they influence early tumorigenesis more than the progression from LGCA to HGCA. This aligns with previous work showing that mRNA levels of CTSB and CTSD increase concurrently with APC gene mutations that initiate the adenoma-adenocarcinoma sequence [50]. This also appeared to be the case for ACE, which was rarely seen except on the endothelium of blood vessels within the NC tissues, and occasionally on the luminal membranes of CA tumor epithelial cells where it was weaker in HGCA than LGCA. The overexpression of PRR mRNA and protein in CA-derived cells reflects its importance in facilitating ERK and Wnt signaling, implicated in CRC initiation. Furthermore, its ability to activate prorenin to initiate the RAS cascade may hint at the potential role of the RAS in these cells.

AT$_1$R is associated with poor cancer outcomes and is therefore of interest when investigating the ways in which the RAS may influence cancer. However, there are currently no specific antibodies against AT$_1$R. One laboratory has tested six commonly-cited anti-AT$_1$R antibodies using various knock-out models and cells known to be negative for AT$_1$R, and found that these antibodies all detected bands at around 43 kDa (the predicted size of AT$_1$R), and produced a variety of staining intensities and localizations by IHC staining not specific for AT$_1$R [48]. AT$_1$R was not detected in our CA tissues or primary cell lines when analyzed by mass spectrometry, possibly because integral membrane proteins can be difficult to isolate during sample preparation [38]. Therefore, despite being a potential target for RAS modulation in the treatment of cancer, determination of AT$_1$R protein expression and localization was not performed in this study.

Overall, IF dual-staining of stemness-associated markers with RAS components in this study and our previous study [27, 28] suggest the presence of a cellular subpopulation that co-expresses NANOG, SOX2, KLF4, c-MYC, CD133, EpCAM, LGR5, AT$_2$R, AEC2, PRR, CTSB, and CTSD. Expression of these RAS components was typically higher in CA tissues than their patient-matched NC tissues. This provides a rationale for targeting CA CSCs with RASIs.

Research investigating cathepsins in CRC has focused on their ability to facilitate cancer cell migration by degrading ECM components, leading to tumor budding, growth factor activation and nutrient recycling [18, 19, 51, 52]. However, angiotensins are also proven substrates for cathepsins, with immune-cell-derived CTSG recently found to have a similar capacity as ACE for converting ATI to ATII [53]. The lack of CTSG expression by CA tumor cells may relate to the finding that it reduces the risk of metastasis by increasing cell-cell adhesion [54]. The presence of active CTSB and CTSD in CA tissues and CA-derived primary cell lines implies that RAS inhibition could be bypassed by these cathepsins, although their precise function remains to be investigated. Therefore, blockade upstream and downstream of the RAS pathway by β-blockers (to reduce pro-renin production) and antagonists of AT$_1$R (ARBs) and AT$_2$R, may mitigate RAS bypass.

The ACE protein was not detected in CA-derived cells by WB or in CA tissues by IF staining, with weak staining of the luminal membrane of CA epithelial cells demonstrated by IHC staining. RT-qPCR detected ACE mRNA, generally at higher abundance in CA tissues than their patient-matched NC samples, but below detectable levels in the CA-derived cells. Despite a lack of effect from ACEIs on the CA-derived primary cell lines, there may be benefits *in vivo* where ACE is transcribed and possibly translated in CA tissues. The ARB losartan did not affect cellular metabolism but reduced *OCT4*, *SOX2* and *NANOG* mRNA expression, and increased *KLF4* expression, relative to untreated cells. This shows that an effect on metabolism does not necessarily accompany a change in gene expression. It would therefore be interesting to assess the effect of ACEIs on the expression of stemness-associated markers. However, WB and RT-qPCR data suggests that the cell lines used in this study may be derived from a cell type devoid of ACE or have lost their expression of ACE, and are therefore an unsuitable model for studying the efficacy of ACEIs in CA. ACEIs have been shown to be beneficial in cancer patients. The liver produces AGT and has high levels of ATII and $AT_1R$. It is the most common metastatic site for CRC, and the metastases contain higher levels of ACE and MasR compared with the primary tumor [16]. The liver may be a fertile ground for CRC metastasis due to its production of AGT, which CRC metastases could utilize via elevated levels of ACE to produce more ATII and drive cancer growth via $AT_1R$-mediated mechanisms. This is supported by the observation that RASIs reduce the spread of CRC to the liver [55, 56], providing a rationale for further investigation into the use of ACEIs in CA.

With the exception of HGCA3, untreated cells expressed detectable levels of *OCT4*, *SOX2* and *NANOG*, which were all downregulated by the administration of RASIs (Fig 10A–10C). Furthermore, while *KLF4* was detected in untreated cells, its expression was significantly lower than in the NC tissue, and cells treated with R-propranolol or EMA401 recovered *KLF4* expression to levels close to that of the NC tissue (Fig 10D). This change in the expression of pluripotency genes was seen at doses that caused minor changes to cell metabolism. These results suggest that RASIs may affect CSCs by attenuating the expression of stemness-associated markers and guiding them away from a pluripotent phenotype. In this case, RAS inhibition may enhance the effect of conventional therapies on CA by making CSCs more susceptible and therefore reducing recurrence and metastasis. It would be interesting to investigate whether CA-derived cells have lower resistance to conventional therapies if they are simultaneously exposed to RASIs, and the effect this might have on the expression of stemness-associated markers and stem cell functionality.

To determine whether any observations were specific to single drugs or a class effect, two drugs from each class were investigated. Propranolol and timolol did not elicit comparable results, suggesting the effects of β-blockers may vary depending on their individual characteristics, such as lipophilicity, membrane stabilizing effect or selectivity for $β_1$, $β_2$ and $α_1$ adrenergic receptors. $β_1$ receptors are most commonly expressed in the heart and kidneys [57]. Second-generation selective β-blockers have a higher affinity for $β_1$ receptors, which are the main target for the treatment of hypertension [58]. First-generation β-blockers have an equal affinity for $β_1$, $β_2$ and $α_1$ receptors and are known as non-selective β-blockers [57]. $β_2$ adrenergic receptors are predominantly expressed in the lungs and gastrointestinal tract [57]. $α_1$ receptors are expressed by endothelial cells and cause vasoconstriction when activated, so selective antagonism of these receptors by third generation β-blockers causes vasodilation and relieves hypertension [21, 57]. This result was interesting because propranolol and timolol are both non-selective β-blockers and renin release is triggered by activation of both $β_1$ and $β_2$ adrenergic receptors [59]. This indicates that the difference in efficacy may be related to adrenergic responses other than renin release, or that the relative potency of propranolol and timolol is the reverse in cell culture compared to an *in vivo* setting due to differences in drug

metabolism. Masur *et al.* [60] found that SW480 cell migration seems to be mediated by $\beta_2$ receptors because selective $\beta_1$-blockers could not prevent migration whereas propranolol could, and propranolol also reduces colon cancer cell proliferation and viability [46, 61]. This indicates that the $\beta_2$ receptor mediates the effects of β-blockers on colon cancer cells, most likely because $\beta_2$ receptors are predominant in the gastrointestinal tract [57]. Therefore, future experiments using $\beta_1$ or $\beta_2$ receptor knock-out cells could reveal whether any effects are mediated by β-adrenergic receptors and, if so, clarify the significance of each receptor type in the response of cancer cells to β-blockers. It would also be interesting to measure renin levels before and after β-blocker treatment to see whether the effects are due to reduced renin and therefore reduced RAS signaling, or due to other β-adrenergic receptor-mediated responses.

Renin can be directly antagonized using aliskiren; however, administration of aliskiren is generally avoided in favor of drugs with fewer side effects and higher bioavailability [62]. Therefore, rather than directly blocking renin, cells were exposed to β-blockers to reduce the production of renin. It is important to keep in mind that because of their wide-ranging physiological significance, antagonism of β-adrenergic receptors may lead to changes in cell metabolism or growth unrelated to renin levels.

Both of the $AT_2R$ antagonists used in this study affected the CA-derived primary cell lines similarly, suggesting that it is a class effect of $AT_2R$ antagonists. $AT_1R$ is internalized and then recycled to the cell membrane when it binds ATII, whereas $AT_2R$ is not internalized [63], therefore the difference in efficacy between EMA401 and SMM02 at 72 h should not be due to the two drugs affecting receptor internalization and recycling speed. However, it could be due to the duration each antagonist is bound to $AT_2R$. One limitation of this experiment was the 72 h time frame, which could be extended in future studies. A longer assay would provide a better idea of whether the drugs continue to affect the cells in the same way over time and resolve the observation of a possible plateau in efficacy or a loss of sensitivity, demonstrated in the drug assays for ARBs (Fig 8M–8P and S6 Fig) and $AT_2R$ antagonists (Fig 8Q–8T and S7 Fig). Furthermore, it would provide the drugs with more time to take effect, particularly on gene and protein expression.

Interestingly, the efficacy of β-blockers and $AT_2R$ antagonists aligns with the hypothesis that cathepsins constitute bypass loops of the RAS and are capable of circumventing ACE inhibition, but that blockade either upstream or downstream of the RAS would not allow cathepsins to act in this way. Accordingly, β-blockers and $AT_2R$ antagonists seemed to have the greatest effects on metabolism, tumorsphere formation and mRNA expression of stemness-associated genes by HGCA-derived cells.

## Conclusions

$AT_2R$, PRR and CTSD are upregulated in CA tissues, whereas ACE and CTSB have similar abundances in CA tissues compared to their patient-matched NC tissues. This suggests that RAS signaling is present within CA to a greater extent than in the NC. IF staining revealed that the NANOG+ CSC-like subpopulation expresses $AT_2R$ and ACE2 and is likely to express PRR, CTSB and CTSD, and that OCT4+ stromal cells express $AT_2R$ which is localized to the nucleus. Although there are smaller impacts on metabolism than expected based on published *in vivo* studies, the changes in gene expression and tumorsphere-forming capability suggest that RASIs may attenuate CSC functionality. Furthermore, the role of $AT_2R$ in CA may be greater than first thought, with EMA401 able to inhibit cancer cell metabolism in a dose-dependent manner, reduce expression of stemness-associated markers and hinder tumorsphere formation. It is yet to be determined whether RASIs may work synergistically with standard chemotherapeutic agents by influencing treatment-resistant CSCs.

## Supporting information

**S1 Fig. Schema demonstrating the classical (black) renin-angiotensin system with enzymes that constitute bypass loops (blue).** Activation of (pro)renin occurs upon binding with (pro) renin receptor. Renin then converts angiotensinogen (AGT) into angiotensin I (ATI), which is cleaved by angiotensin-converting enzyme (ACE) to produce the active peptide angiotensin II (ATII). Cathepsin B and cathepsin D contribute to renin activation. Cathepsin D and chymase mediate conversion of AGT into ATI. Cathepsin G promotes generation of ATII from ATI or directly from AGT. Angiotensin III (ATIII) is the result of further cleavage of ATII by amino-peptidase A. Angiotensin IV (ATIV) is formed by cleavage of ATIII by aminopeptidase N. Both ATII and ATIII act on angiotensin II receptor 1 ($AT_1R$) and angiotensin II receptor 2 ($AT_2R$). ACE2 converts ATI and ATII into Ang1-9 and Ang1-7, which both bind the Mas receptor. The downstream effects of $AT_2R$ and Mas receptor agonism counteract those of $AT_1R$. Adapted with permission from Munro et al. *Integr Cancer Sci Therap.* [10] and from [38] under a CC BY license, with permission from Munro MJ *et al.* 2021.
(JPG)

**S2 Fig. Cathepsin activity assays.** Activity assays were performed for cathepsin B in colon adenocarcinoma (CA) tissues (A) and cells (C), and for cathepsin D in CA tissues (B) and CA-derived cells (D). Fluorescence intensity units (FIU) were divided by the total amount of protein added to give a measure of activity per μg of protein. The average FUI per μg of protein across 2 technical replicates is displayed, with error bars representing standard deviation. Reproduced from [38] under a CC BY license, with permission from Munro MJ *et al.* 2021.
(JPG)

**S3 Fig. Effect of propranolol on the metabolism of HGCA-derived primary cell lines.** Four HGCA-derived cells lines were exposed to R-propranolol (A-D), S-propranolol (E-H) and racemic propranolol (I-L) at doses of 50 μM (●), 10 μM (■) and 1 μM (▲). Luminescence produced by cells exposed to propranolol was detected and used to measure the inhibitory effect of treatment on metabolic activity. The y-axis represents the extent of metabolic inhibition in the treated cells relative to the untreated control cells, and because the assay seeks to measure metabolic inhibition, the positive values indicate inhibition and the negative values indicate increased metabolic activity in the drug-treated cells. The x-axis shows the time after initial dose of each luminescence measurement. Each data point is the average of 3 technical repli-cates, with error bars showing standard deviation. Reproduced from [38] under a CC BY license, with permission from Munro MJ *et al.* 2021.
(JPG)

**S4 Fig. Effect of timolol on the metabolism of HGCA-derived primary cell lines.** Four HGCA-derived cells lines were exposed to R-timolol (A-D) and S-timolol (E-H) at doses of 100 μM (●), 50 μM (■) and 10 μM (▲). Luminescence produced by cells exposed to proprano-lol was detected and used to measure the inhibitory effect of treatment on metabolic activity. The assay was developed to measure metabolic inhibition, and so the positive values on the y-axis represent a positive outcome in terms of inhibiting metabolism. Therefore, positive values show the extent of metabolic inhibition in the treated cells relative to the untreated control cells, and the negative values indicate increased metabolic activity in the drug-treated cells. The x-axis shows the time after initial dose of each luminescence measurement. Each data point is the average of 3 technical replicates, with error bars showing standard deviation. Reproduced from [38] under a CC BY license, with permission from Munro MJ *et al.* 2021.
(JPG)

**S5 Fig. Effect of ACEIs on the metabolism of HGCA-derived primary cell lines.** Four HGCA-derived cells lines were exposed to captopril (A-D) and cilazapril (E-H) at doses of 100 μM (●), 50 μM (■) and 10 μM (▲). Luminescence produced by cells exposed to ACEIs was detected and used to measure the inhibitory effect of treatment on metabolic activity. The assay was developed to measure metabolic inhibition, and so the positive values on the y-axis represent a positive outcome in terms of inhibiting metabolism. Therefore, positive values show the extent of metabolic inhibition in the treated cells relative to the untreated control cells, and the negative values indicate increased metabolic activity in the drug-treated cells. The x-axis shows the time after initial dose of each luminescence measurement. Each data point is the average of 3 technical replicates, with error bars showing standard deviation. Reproduced from [38] under a CC BY license, with permission from Munro MJ *et al.* 2021. (JPG)

**S6 Fig. Effect of angiotensin receptor blockers (ARBs) on the metabolism of HGCA-derived cell lines.** Four HGCA-derived cells lines were exposed to candesartan (A-D) and losartan (E-H) at doses of 100 μM (●), 50 μM (■) and 10 μM (▲). Luminescence produced by cells exposed to ARBs was detected and used to measure the inhibitory effect of treatment on metabolic activity. The assay was developed to measure metabolic inhibition, and so the positive values on the y-axis represent a positive outcome in terms of inhibiting metabolism. Therefore, positive values show the extent of metabolic inhibition in the treated cells relative to the untreated control cells, and the negative values indicate increased metabolic activity in the drug-treated cells. The x-axis shows the time after initial dose of each luminescence measurement. Each data point is the average of 3 technical replicates, with error bars showing standard deviation. Reproduced from [38] under a CC BY license, with permission from Munro MJ *et al.* 2021. (JPG)

**S7 Fig. Effect of AT$_2$R antagonists on the metabolism of HGCA-derived primary cell lines.** Four HGCA-derived cells lines were exposed to EMA401 (A-D) and SMM02 (E-H) at doses of 100 μM (●), 50 μM (■) and 10 μM (▲). Luminescence produced by cells exposed to AT$_2$R antagonists was detected and used to measure the inhibitory effect of treatment on metabolic activity. The assay was developed to measure metabolic inhibition, and so the positive values on the y-axis represent a positive outcome in terms of inhibiting metabolism. Therefore, positive values show the extent of metabolic inhibition in the treated cells relative to the untreated control cells, and the negative values indicate increased metabolic activity in the drug-treated cells. The x-axis shows the time after initial dose of each luminescence measurement. Each data point is the average of 3 technical replicates, with error bars showing standard deviation. Reproduced from [38] under a CC BY license, with permission from Munro MJ *et al.* 2021. (JPG)

## Acknowledgments

The authors would like to thank Ms Liz Jones for her assistance with IHC and IF staining, and Dr John Groom and colleagues at Hutt Hospital and Wakefield Hospital for facilitating recruitment of tissue samples from their patients to the GMRI Tissue Bank. We thank Dr Sean Mackay of University of Otago for providing the proprietary AT$_2$R antagonist SMM02.

## Author Contributions

**Conceptualization:** Matthew J. Munro, Lifeng Peng, Susrutha K. Wickremesekera, Swee T. Tan.

**Data curation:** Matthew J. Munro.

**Formal analysis:** Matthew J. Munro.

**Funding acquisition:** Swee T. Tan.

**Investigation:** Matthew J. Munro.

**Methodology:** Matthew J. Munro.

**Project administration:** Swee T. Tan.

**Resources:** Swee T. Tan.

**Supervision:** Lifeng Peng, Susrutha K. Wickremesekera, Swee T. Tan.

**Validation:** Matthew J. Munro, Lifeng Peng, Swee T. Tan.

**Visualization:** Lifeng Peng, Swee T. Tan.

**Writing – original draft:** Matthew J. Munro.

**Writing – review & editing:** Matthew J. Munro, Lifeng Peng, Susrutha K. Wickremesekera, Swee T. Tan.

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
