## [Decision Letter · Decision Letter 0]

13 Jul 2021

PONE-D-21-19576

Colon adenocarcinoma-derived cells possessing stem cell function can be modulated using renin-angiotensin system inhibitors

PLOS ONE

Dear Dr. Tan,

Thank you for submitting your manuscript to PLOS ONE. After careful consideration, we feel that it has merit but does not fully meet PLOS ONE’s publication criteria as it currently stands. Therefore, we invite you to submit a revised version of the manuscript that addresses the points raised during the review process.

Interesting and new topic. The paper is too long mainly in the Introduction. Figures are too much  and must put together in panels or deleted. Some of the  figures are badly stained or out of focus, some data are missing and figures must be put in the correct order. 

We look forward to receiving your revised manuscript.

Kind regards,

Gianpaolo Papaccio, M.D., Ph.D.

Academic Editor

PLOS ONE

Additional Editor Comments (if provided):

This is an interesting study/perspective on renin-angiotensin system inhibitors in CRC stem cells.

The aim and topic are new, the experiments were well designed though the Figures are sometimes out of focus or not well stained.

On the other hans they are excessive so that several must be deleted (the badly stained in particular.

Moreover the length of the Introduction is also excessive and must be cut. It is much more long than the Discussion.

Therefore the Authors must put together some figures in panels and delete the bad ones, and put all figures in the correct order with respect to paragraphs.

Then reduce the Introduction to a third at least (remember that less is more).

Finally add missing data like those in figure 5.

Journal Requirements:

"This project is supported by the Lloyd Morrison Trust and MM was supported by a PhD Scholarship from the New Zealand Community Trust."

"The authors would like to thank Ms Liz Jones for her assistance with IHC and IF staining, and Dr John Groom and colleagues at Hutt Hospital and Wakefield Hospital for their support in facilitating recruitment of tissue samples from their patients to the GMRI Tissue Bank. This project is supported by the Lloyd Morrison Trust and MM was supported by a PhD Scholarship from the New Zealand Community Trust."

"This project is supported by the Lloyd Morrison Trust and MM was supported by a PhD Scholarship from the New Zealand Community Trust."

5. We note that you have a patent relating to material pertinent to this article. Please provide an amended statement of Competing Interests to declare this patent (with details including name and number), along with any other relevant declarations relating to employment, consultancy, patents, products in development or modified products etc. Please confirm that this does not alter your adherence to all PLOS ONE policies on sharing data and materials, as detailed online in our guide for authors http://journals.plos.org/plosone/s/competing-interests by including the following statement: "This does not alter our adherence to  PLOS ONE policies on sharing data and materials.” If there are restrictions on sharing of data and/or materials, please state these. Please note that we cannot proceed with consideration of your article until this information has been declared.

6. We note that Figures 1, 2, 3, 6, 7, 8, 9, 10, 11, and 17 in your submission contain copyrighted images. All PLOS content is published under the Creative Commons Attribution License (CC BY 4.0), which means that the manuscript, images, and Supporting Information files will be freely available online, and any third party is permitted to access, download, copy, distribute, and use these materials in any way, even commercially, with proper attribution. For more information, see our copyright guidelines: http://journals.plos.org/plosone/s/licenses-and-copyright.

a. You may seek permission from the original copyright holder of Figures 1, 2, 3, 6, 7, 8, 9, 10, 11, and 17 to publish the content specifically under the CC BY 4.0 license. 

Reviewers' comments:

Reviewer's Responses to Questions

**Comments to the Author**

1. Is the manuscript technically sound, and do the data support the conclusions?

Reviewer #1: Yes

2. Has the statistical analysis been performed appropriately and rigorously? 

Reviewer #1: Yes

3. Have the authors made all data underlying the findings in their manuscript fully available?

Reviewer #1: Yes

4. Is the manuscript presented in an intelligible fashion and written in standard English?

Reviewer #1: Yes

5. Review Comments to the Author

Reviewer #1: Authors present an interesting perspective on renin-angiotensin system inhibitors in CRC stem cells. The topic is quite new, experiments are well designed and conducted. Nevertheless, too many figures and the excess of words make this manuscript difficult to read. Figures can be put together so that at each result paragraph correspond one figure only. this also with respect to figure order, which does not follow results order. Some data are missing in some figures, such as figure 5.

6. PLOS authors have the option to publish the peer review history of their article (what does this mean?). If published, this will include your full peer review and any attached files.

Reviewer #1: No

---

## [Author Response · Author response to Decision Letter 0]

3 Aug 2021

We thank you and the reviewer for your support and suggestions. We have taken on board the helpful comments and have revised our manuscript accordingly. We have reduced the word count including substantially shortening the Introduction section and reducing the number of figures (from 18 to 10) by creating multi-panel figures and presenting some of the data as additional supplementary figures. We trust that this is now acceptable.

---

## [Editor Report · Decision Letter 1]

4 Aug 2021

Colon adenocarcinoma-derived cells possessing stem cell function can be modulated using renin-angiotensin system inhibitors

PONE-D-21-19576R1

Dear Dr. Tan,

We’re pleased to inform you that your manuscript has been judged scientifically suitable for publication and will be formally accepted for publication once it meets all outstanding technical requirements.

Kind regards,

Gianpaolo Papaccio, M.D., Ph.D.

Academic Editor

PLOS ONE

Additional Editor Comments (optional):

The Authors answered to all previous criticisms.
---

## [Editor Report · Acceptance letter]

13 Aug 2021

PONE-D-21-19576R1 

Colon adenocarcinoma-derived cells possessing stem cell function can be modulated using renin-angiotensin system inhibitors 

Dear Dr. Tan:

I'm pleased to inform you that your manuscript has been deemed suitable for publication in PLOS ONE. Congratulations! Your manuscript is now with our production department. 

Kind regards, 

on behalf of

Prof. Gianpaolo Papaccio 

Academic Editor

PLOS ONE